# Model Predictive Control of a Semi-Active Vehicle-Mounted Vibration Isolation Platform

**DOI:** 10.3390/s24010243

**Published:** 2023-12-31

**Authors:** Liang Wu, Weizhou Zhang, Daofa Yuan, Iljoong Youn, Weiwei Jia

**Affiliations:** 1State Key Laboratory of Automotive Simulation and Control, Jilin University, Changchun 130012, China; astdwxg@jlu.edu.cn (L.W.); wzzhang21@mails.jlu.edu.cn (W.Z.); yuandaofa@huawei.com (D.Y.); 2Changsha Automotive Innovation Research Institute, Changsha 410036, China; 3School of Mechanical and Aerospace Engineering, Gyeongsang National University, Jinju 660-701, Republic of Korea; iyoun@gnu.ac.kr; 4School of Management Science and Information Engineering, Jilin University of Finance and Economics, Changchun 130117, China

**Keywords:** semi-active vehicle-mounted vibration isolation platform, input constraints, road preview, model predictive control, vibration environment test

## Abstract

When conventional delivery vehicles are driven over complex terrain, large vibrations can seriously affect vehicle-loaded equipment and cargo. Semi-active vehicle-mounted vibration isolation control based on road preview can improve the stability of loaded cargo and instruments by enabling them to have lower vertical acceleration. A combined dynamic model including a vehicle and platform is developed first. In order to obtain a non-linear relationship between damping force and input current, a continuous damping control damper model is developed, and the corresponding external characteristic tests are carried out. Because some conventional control algorithms cannot handle complex constraints and preview information, a model predictive control algorithm based on forward road preview and input constraints is designed. Finally, simulations and real tests of the whole vehicle vibration environment are carried out. The results show that the proposed model predictive control based on road preview can effectively improve vibration isolation performance of the vehicle-mounted platform.

## 1. Introduction

With the progress of science and technology, aerospace and geological exploration and navigation have played an increasingly significant role in society’s progress. In these fields, high-precision ground detection equipment is implemented to detect underwater mineral and oil resources. Aerial detection can cover a wide area, but underground detail may be lost due to high speed and distance from the ground. Vehicle-loaded detection can be applied to collect more detailed underground information.

Ground environments are usually complex, including mountains and desert, and they experiencing extreme weather. Vibrations caused by harsh roads undoubtedly have a big impact on equipment, interfering with the measurement accuracy. In the suspension design of the loaded vehicle, a trade-off between ride comfort and handling capability must be made. The effect of unexpected accelerations on the upper detection equipment cannot be expected to be entirely eliminated by the vehicle suspension.

Due to the high randomness of the ground surface and limitations of the vehicle suspension design, a vibration isolation system acting on the equipment must be designed to reduce the effect of unexpected accelerations and maintain measurement accuracy and transportation safety [1,2].

In ground vehicles, active or semi-active suspensions have been utilized to solve the problems of vibrations for several decades. For eliminating vertical vibrations, which come from rough roads, an active suspension provides up and down forces to reduce both the accelerations of sprung mass and tire defections. In this way, the ride comfort and handling capability can be improved.

In recent years, many scholars have conducted research on innovative suspensions, such as mechatronic [3,4], pneumatic [5], hydraulic [6], and mixed [7] suspensions. To achieve a balance between comfort and maneuverability, a nonlinear control law based on nonlinear hydraulic actuator-regulating hydraulic valves is used to provide active control force. In the presence of sensor noise and parameter uncertainty, the method is robust and has low hydraulic power consumption [8]. An adaptive neural network inverse step control method is applied to a nonlinear electro-hydraulic active suspension control system to explicitly tackle the trade-off between passenger comfort/road handling and passenger comfort/suspension [9]. For solving the problem of controller parameter selection, a variety of controller parameter optimization methods based on artificial intelligence are used to improve vehicle driving comfort and operating stability. The results show that the fuzzy LQR control method optimized by particle swarm optimization (PSO) can achieve better results in body vibration, vehicle acceleration, suspension deflection, and tire deflection [10].

The active suspensions can obviously improve ride comfort by the reduction of vertical accelerations, but the ride comfort also can be affected by accelerations from lateral and longitudinal directions, especially on bumpy roads or ramp roads. A more active attitude toward motions of the car body is used to eliminate the unpleasant lateral and longitudinal accelerations acting on passengers. Therefore, the vibrations in vertical, longitudinal, and lateral directions can be reduced simultaneously [11,12]. However, the energy consumption problems still limit the application of active suspensions. Semi-active suspensions are a more economical solution. They provide limited performance compared to active suspensions due to the limitations of force characteristics [13] and the absence of a power source. However, semi-active suspensions offer significant cost and weight advantages.

A new adaptive semi-active suspension control method is proposed. The trade-off between comfort and stability is realized by combining vehicle speed, road data, and forward road data prediction. By modifying the controller under the LPV framework, the control reconfiguration is realized, and the passenger comfort and vehicle stability are improved [14]. A robust control method using an interconnected air suspension system based on mode switching is proposed to adjust the dynamic performance of the air suspension system. Under different driving conditions, the ride comfort and operation stability can be effectively improved through the mode switching strategy [15].

A new method for vehicle dynamic control by semi-active and active suspension systems is proposed. Considering tire nonlinearity, supporting force, and suspension geometry resistance characteristics, a nonlinear model predictive control algorithm is used to effectively control vehicle pitch and roll. For the Audi E-Tron prototype, the CVSA2-Kinetic continuously variable semi-active suspension system is adopted to conduct simulation research, including a virtual driver model, driving force layer, reference generator, and vehicle stability controller. The results show that the system can effectively improve the driving stability and comfort of the vehicle [16].

It is imperative to design a semi-active vehicle-mounted vibration isolation platform (VVP) according to the requirements. It is commonly used for the vibration protection of detection equipment such as measuring instruments and optical instruments [17]. The structure of the vibration isolation platform is composed of three main parts: the upper plane, base, and intermediate vibration isolation system. An air spring and two semi-active dampers are arranged at one of four corners, as shown in Figure 1.

In previous research, the parameters of the damping system are optimized by NSGA_II and MOPSO to significantly improve the anti-vibration performance of the VVP. The sine excitation is verified by experiments, and the error between experiments and simulation is as small as 8.30%, thus verifying the effectiveness of the optimization parameters [18]. The complex mechanical structure was simplified as a 5-DOF linear model for LQR controller design. The semi-active control force can be obtained by changing the inertance coefficient in order to track the active control force in step. The performance of the control system has been investigated for C-class random road disturbances for the simulated vehicle system [19].

Although classical linear control algorithms, such as LQR, can improve the vibration isolation performance of the VVP, there are nonlinear physical constraints on the semi-active continuous damping control (CDC) damper. The traditional linear control algorithm ignores the nonlinear input constraints in the solution process. Therefore, the absence of nonlinear constraints in the semi-active control system may lead to a significant reduction in the optimization effect in practical applications [20].

In recent years, many scholars have studied the model predictive control (MPC) on suspensions, considering road preview [21,22,23]. Road preview helps improve control performance by inputting the road spectrum information ahead of the controller in advance. The controller can plan the optimal or near-optimal control action based on this information before the arrival of the road input. In addition, the development of technologies such as sensors and microprocessors has made the application of road preview in vehicles a reality. The performance of the MPC controller is investigated and compared with non-linear constraints, and then a novel method is proposed for sensors to acquire road preview information [24,25]. One of the limiting factors for the widespread adoption of MPC in the industry is that the traditional implicit MPC can impose a huge computational load on the controller during computation because of the optimized calculation for each sampling interval. To address this problem, we propose a suspension system with an explicit Model Predictive Controller (e-MPC), which runs the MPC optimization process offline and simplifies the online controller for functional evaluation, reducing memory usage [26].

Based on the above analysis and the previous research background, an MPC algorithm with prospective road information is applied to the control of the VVP. The physical constraints of the CDC damper are considered in the solution process to reduce the vertical accelerations of the VVP. The designed MPC control algorithm is verified and comparatively analyzed in terms of vibration isolation performance through simulations and tests.

## 2. Materials and Methods

As shown in Figure 2, when the vehicle is in motion, the vibration excitation of the VVP is mainly from the ground. The vibration is transmitted from the ground, wheels, vehicle suspension, and VVP to the precision instrument. The VVP is designed to attenuate the vibration transmitted from the body to the precision instrument. In order to study the dynamics of the VVP, an 8 degree-of-freedom (DOF) dynamic model combining the vehicle and the platform is first built in this section. A CDC damper is modelled based on test data. In order to simulate the vibration response of the system, three road surface performance models are established. The parameters of the VVP and vehicle are shown in Table 1.

### 2.1. Dynamic Modelling of a Combined Vehicle and Platform

In order to apply the MPC control strategy to the VVP, an 8-DOF mathematical model combining the VVP and the mounted vehicle is built as shown in Figure 3. The following assumptions are made in the modelling process: (1) The VVP is fixed to the vehicle body by the base. The mass of the base is included in the body mass. (2) To facilitate the study of the vibration isolation performance of the VVP in the vertical direction, the vehicle is considered to be moving at constant speed in a straight line, and the slip between the wheels and the ground is neglected. (3) The main vibration of the VVP is from the vertical direction. The car body moves in the vertical direction, rolling and pitching.

The equation for the vertical motion of the VVP is indicated as
(1)$$m_{p}{\ddot{z}}_{p}=k_{p}\left(z_{c}+L_{t}\varphi +L_{p}\theta -z_{p}\right)+\left(c_{p}+c_{sem}\right)\left({\dot{z}}_{c}+L_{t}\dot{\varphi }+L_{p}\dot{\theta }-{\dot{z}}_{p}\right)$$
where $z_{c}$ and $z_{p}$ are the vertical displacement of the mass center of the body and the mass center of the VVP, respectively; and θ and φ are the pitch and roll angles of the car body, respectively.

The vertical dynamic equation of the car body is expressed as
(2)$$M{\ddot{z}}_{c}=\sum_{i=1}^{4}f_{i}-m_{p}{\ddot{z}}_{p}$$
where $f_{i}$ is the forces acting on car body by the suspension.

The rolling dynamic equation of the car body is expressed as
(3)$$I_{r}\ddot{\varphi }=\frac{T_{f}}{2}\left(f_{1}-f_{2}\right)+\frac{T_{r}}{2}\left(f_{3}-f_{4}\right)+f_{p}L_{t}+Mgh_{r}\varphi$$

The pitching dynamic equation of the body is expressed as
(4)$$I_{p}\ddot{\theta }=-L_{f}\left(f_{1}+f_{2}\right)+L_{r}\left(f_{3}+f_{4}\right)+f_{p}L_{p}+Mgh_{p}\theta$$

The elastic deformations of the vehicle suspensions are expressed as
(5)$$\begin{array}{c} z_{s1}=z_{c}+\frac{T_{f}}{2}\varphi -L_{f}\theta -z_{1}\\ z_{s2}=z_{c}-\frac{T_{f}}{2}\varphi -L_{f}\theta -z_{2}\\ z_{s3}=z_{c}+\frac{T_{r}}{2}\varphi +L_{r}\theta -z_{3}\\ z_{s4}=z_{c}-\frac{T_{r}}{2}\varphi +L_{r}\theta -z_{4} \end{array}$$
where $z_{i}$ is the vertical displacement of the mass center of the four wheels, and $z_{si}$ is the elastic deformation of the vehicle suspension.

The forces acting on the car body by the suspension can be calculated according to Equation (5) as follows:(6)$$f_{i}=-k_{si}z_{si}-c_{i}{\dot{z}}_{si},\;i=1,\;2,\;3,\;4$$

The vertical dynamics equations of the unsprung masses of the vehicle are expressed as
(7)$$m_{i}{\ddot{z}}_{i}=-f_{i}+k_{ti}\left(w_{i}-z_{i}\right),\;i=1,\;2,\;3,\;4$$
where $w_{i}$ is the road excitation of the four wheels.

According to the above formulas, the state vector x, control force vector u, road input vector w, and output vector y of the VVP system can be expressed as
(8)$$\begin{array}{c} x={\left[z_{1},z_{2},z_{3},z_{4},z_{c},z_{p},\varphi ,\theta ,{\dot{z}}_{1},{\dot{z}}_{2},{\dot{z}}_{3},{\dot{z}}_{4},{\dot{z}}_{c},{\dot{z}}_{p},\dot{\varphi },\dot{\theta }\right]}^{T}\\ w={\left[w_{1},w_{2},w_{3},w_{4}\right]}^{T}\\ y={\left[z_{p}-z_{c},{\ddot{z}}_{p}\right]}^{T}\\ u=f_{p} \end{array}$$

The dynamic equations combining vehicle and platform are expressed as follows:(9)$$\begin{array}{c} \dot{x}=Ax+Bu+Ew\\ y=Cx \end{array}$$

### 2.2. CDC Damper Modeling According to Test Data

In order to facilitate more accurate damping force control of the CDC damper, a CDC damper model is established based on the test data of the external characteristics of the damper. According to the known speed and desired damping force, the model is used to invert the input current of the damper at the present time and to determine the damping force boundary value of the CDC damper.

The test data are fitted by the smoothing spline method [27] to obtain the variation curves of damping force and velocity for different input currents. The fitted curves are obtained by minimizing the following equations:(10)$$RRS\left(f,\lambda \right)=\sum_{i=1}^{N}{\left(y_{i}-f\left(x_{i}\right)\right)}^{2}+\lambda \int \left(f^{\prime \prime }\left(t\right)\right)dt$$
where λ is the smoothing coefficient. The magnitude of $\sum_{i=1}^{N}{\left(y_{i}-f\left(x_{i}\right)\right)}^{2}$ indicates how similar the fitted curve is to the original data, and the magnitude of $\lambda \int \left(f^{\prime \prime }\left(t\right)\right)dt$ indicates how smooth the curve is. $\lambda \int \left(f^{\prime \prime }\left(t\right)\right)dt$ is also called the curvature penalty. When λ tends toward 0, the fitted curve can be infinitely bent through all points of the original data. The curve is smoother when λ tends toward ∞.

Through several fitting simulations, a fitting function $f\left(x\right)$ that meets the requirements is finally determined. The relationship curves between damping force, velocity, and current are obtained, as shown in Figure 4.

According to Figure 4, the three-dimensional MAP diagram of damping force, velocity, and current can be obtained by applying the cubic spline interpolation method [28], as shown in Figure 5. The optimal damping force is calculated based on the system state data obtained from the sensors and the MPC control algorithm designed below. The required input current for the CDC damper is calculated according to Figure 5.

### 2.3. Modeling of Road Surface Performance

For VVP simulation and real test analysis, C-class random road surface, discrete impact road surface, and sinusoidal road surface models are established in this section.

#### 2.3.1. C-Class Random Road Surface Performance Model

In order to simulate the vibration response of the VVP under real road conditions using the pavement spectrum correlation theory [29], a C-class random road surface performance model is established using the harmonic superposition method [30,31]. In this paper, the C-class random road surface model is established by superimposing a series of sinusoidal waves.

The variance of the C-class road surface spectrum can be expressed as
(11)$$\sigma^{2}=\int_{f_{1}}^{f_{2}}G_{q}\left(f\right)df$$
where the central frequency of the subinterval is denoted as $P_{mid\_i}$. The integral is discretized [32] by dividing $f_{1}$ to $f_{2}$ into n intervals.

When n tends toward ∞, the power spectral density at the frequency of $P_{mid\_i}$ can be considered as the average power spectral density of the subinterval, so that (11) can be expressed as
(12)$$\sigma^{2}=\sum_{i=1}^{n}G_{q}\left(P_{mid\_i}\right)∆f_{i}$$

According to Parseval’s theorem, at a frequency of $P_{mid\_i}$ and a standard deviation of $\sigma_{i}$, the associated sine wave equation can be expressed as $\sqrt{2}\sigma_{i}\sin\left(2\pi P_{mid\_i}x+\theta_{i}\right)$. Therefore, the random road surface height by superimposing n sine waves can be calculated as
(13)$$f\left(x\right)=\sum_{i=1}^{n}\sqrt{2G_{q}\left(P_{mid\_i}\right)∆f_{i}}\sin\left(2\pi P_{mid\_i}x+\theta_{i}\right)$$

In the simulation analysis, the vehicle is assumed to travel in a straight line at a constant speed of 20 km/h. Therefore, the road surface height with respect to time t can be expressed as
(14)$$f\left(t\right)=\sum_{i=1}^{n}\sqrt{2G_{q}\left(P_{mid\_i}\right)∆f_{i}}\sin\left(2\pi P_{mid\_i}ut+\theta_{i}\right)$$
where the geometric mean value of the road roughness coefficient of the C-class random road surface is $256\times 10^{-6}{\;m}^{3}$, and the geometric mean value of the root mean square of the road roughness coefficient is $15.23\times 10^{-2}\;m$. Since the wheelbase of the mounted vehicle is 4.9 m, the rear wheel input has a lag of 0.882 s relative to the front wheel input.

To better reflect the track of wheels on the C-class random road surface, the Butterworth low-pass filter is used to filter the C-class random road surface. We filter out the high-frequency parts of the pavement that have little contact with the tire [33,34], as shown in Figure 6.

#### 2.3.2. Discrete Impact Road Surface Performance Model

The discrete impact road surface causes instantaneous and large vibration of the vehicle, which is a special working condition of the road surface excitation. Therefore, two discrete impact road surface performance models, including impulse road surface and step road surface, are developed to simulate the vibration response of vehicles passing through speed bumps or manhole covers.

Impulse road surface performance model

The impulse road surface mainly reflects the vibration response of the vehicle when passing over the speed bump. According to GB/T4970-2009 “Vehicle Ride Comfort Test Method”, the mathematical model of the impulse road surface is established as follows:(15)$$w\left(t\right)=\left\{\begin{array}{cc} 0, & 0\leq t\leq \frac{l}{v}\\ \frac{H_{L}}{2}\left(1-\cos\left(\frac{2\pi v}{L}\left(t-\frac{l}{v}\right)\right)\right), & \frac{l}{v}<t\leq \frac{l+L}{v}\\ 0, & t>\frac{l+L}{v} \end{array}\right.$$
where $w\left(t\right)$ is the height of the road surface, l is the distance between the front wheels and the speed bump in the initial state, v is the travel speed of the vehicle, $H_{L}$ is the height of the speed bump, 0.05 m, and L is the width of the speed bump, 0.32 m.

When the vehicle speed is 20 km/h, the impulse road surface performance model is established according to MATLAB as shown in Figure 7.

2.Step road surface performance model

The step road surface mainly reflects the vibration response of the vehicle when passing over a manhole cover. In this section, the step road surface model is established by iterative fitting of the Fourier series.

The trigonometric form of the Fourier series is expressed as
(16)$$f\left(t\right)=\frac{a_{0}}{2}+\sum_{k=1}^{\infty }\left[a_{k}\cos\left(k\omega_{0}t\right)+b_{k}\sin\left(k\omega_{0}t\right)\right]$$

The ideal road surface input in a period T is expressed as
(17)$$f\left(t\right)=\left\{\begin{array}{cc} \frac{E}{2}, & 0\leq t\leq \frac{T}{2}\\ -\frac{E}{2}, & \frac{T}{2}<t<T\; \end{array}\right.$$
where E is the pavement disturbance amplitude.

Based on the orthogonality of the trigonometric functions, the coefficients of the Fourier series, $a_{0}$, $a_{k}$, and $b_{k}$ are calculated as
(18)$$\begin{array}{l} a_{0}=\frac{2}{T}\int_{0}^{T}f\left(t\right)dt=0\\ a_{k}=\frac{2}{T}\int_{0}^{T}f\left(t\right)\cos\left(k\omega_{0}t\right)dt=0\\ b_{k}=\frac{2}{T}\int_{0}^{T}f\left(t\right)\sin\left(k\omega_{0}t\right)dt=\frac{2E}{Tk\omega_{0}}\left(1-\cos\left(\frac{k\omega_{0}T}{2}\right)\right) \end{array}$$
where T=5π, $\omega_{0}=0.4$, and $\frac{E}{2}=0.04$

Substituting Equation (18) into (16), the expansion of the Fourier series can be obtained as
(19)$$f\left(t\right)=\sum_{k=1}^{\infty }\frac{2E}{\left(2k-1\right)\pi }\sin\left(0.4\left(2k-1\right)t\right)$$

When the vehicle speed is 20 km/h, the step road surface performance model is established according to MATLAB, as shown in Figure 8.

#### 2.3.3. Sinusoidal Road Surface Performance Model

In this section, two road surface models, including sinusoidal fixed frequency road surface and sinusoidal swept frequency road surface, are developed to simulate the vibration response of the VVP when the vehicle is subjected to continuous impact.

Sinusoidal fixed frequency road surface performance model

According to the requirements of simulations and real tests, the sinusoidal fixed frequency road surface model with a frequency of 3.6 Hz and amplitude of 0.05 m is established, as shown in Figure 9.

2.Sinusoidal swept frequency road surface performance model

The frequency domain response of the VVP can be obtained by sweeping the road surface, which is convenient for finding the optimal vibration isolation interval and the resonance peak. In this simulation and real test process, a sweep spectrum with a linear increase in acceleration excitation with time is established; the sweep frequency range is 0.2 to 6 Hz, with a total sweep time of 60 s. The sinusoidal swept frequency road surface model is
(20)$$\left\{\begin{array}{l} y=\frac{100acc}{4\pi^{2}t^{2}}\sin\left(2b\pi t^{2}\right),\;t_{min\;}\leq t\leq t_{max\;}\\ acc=\left(1+\frac{1}{12}t\right),\;t_{min\;}\leq t\leq t_{max\;} \end{array}\right.$$
where acc is the acceleration amplitude with an initial value of 1 m/s^2^, which increases linearly with time and increases to 6 m/s^2^ when the sweep time reaches 60 s; b is the sweep coefficient, with a value of 0.1, which indicates how fast the frequency changes with time. The sinusoidal swept frequency road surface performance model is established according to MATLAB, as shown in Figure 10.

## 3. Results for Design of Control Algorithms for VVP

In this section, the nonlinear factors of the CDC damper are constrained. The road preview information is processed and analyzed. The MPC control algorithm for vibration isolation control is designed considering nonlinear constraints.

In the control of the CDC damper-based VVP, the vertical acceleration of the VVP, platform dynamic deflection, and CDC damper input are considered as performance indicators. Therefore, the cost function of vibration isolation control of the VVP can be expressed as
(21)$$J=\rho_{1}{\ddot{z}}_{p}^{2}+\rho_{2}{\left(z_{p}-z_{c}\right)}^{2}+\rho_{3}u^{2}$$
where $\rho_{1}$, $\rho_{2}$, and $\rho_{3}$ are the vertical acceleration of the VVP, platform dynamic deflection, and CDC damper input weight coefficients, respectively.

### 3.1. CDC Damper Nonlinear Constraint Treatment

The adjustable damping force of the CDC damper in the VVP system is
(22)$$u=c_{sem}\left({\dot{z}}_{p}-{\dot{z}}_{c}\right)+F_{0}sign\left({\dot{z}}_{p}-{\dot{z}}_{c}\right)$$
where the damping factor $c_{sem}$ is determined by the area of the CDC damper solenoid valve opening.

The nonlinear relationship between damping force, velocity, and current for the CDC damper is shown in Figure 4. To facilitate the application of input constraints to the VVP controller designed below, the nonlinear relationship between damping force, velocity, and current is simplified by retaining the maximum and minimum curves of the damping force available at each velocity and linearizing the curve segments to obtain the constrained interval of the CDC damper damping force, as shown in Figure 11. The shaded area in Figure 11 limits the actual values that can be obtained for the CDC damper damping force u. $u_{min}\left(v_{df}\right)$ and $u_{max}\left(v_{df}\right)$ are the minimum and maximum boundaries for the values that can be taken for the damping force u, respectively, where the linearized boundaries can be expressed as
(23)$$u=\alpha_{i}v_{df}+\beta_{i},\;i=1,\;2,\;3,\;4,\;5$$
where $\alpha_{i}$ and $\beta_{i}$ are the parameter terms that limit the adjustable damping force to the shaded region. The optimal control damping force for solving the semi-active MPC control strategy for the VVP must satisfy the following inequality constraint:(24)$$\left\{\begin{array}{c} u\leq \alpha_{4}v_{df}+\beta_{4}\\ u\leq \alpha_{5}v_{df}+\beta_{5}\\ u\geq \alpha_{3}v_{df}+\beta_{3} \end{array}\right.v_{df}\geq 0;\left\{\begin{array}{c} u\leq \alpha_{2}v_{df}+\beta_{2}\\ u\geq \alpha_{3}v_{df}+\beta_{3}\\ u\geq \alpha_{5}v_{df}+\beta_{5} \end{array}\right.v_{df}\leq 0$$

### 3.2. Analysis and Processing of Road Surface Preview Information

By transforming the obtained road surface height preview information into road surface height matrices under rolling time series, the matrix is combined with the system state space equations and applied to the design of the MPC controller to improve the performance of the controller. Assume that $\hat{w}\left(k\right)$ is the height information of the road ahead with the preview interval $\left[0,N_{p}\right]$ at moment *k*. Therefore, $\hat{w}\left(k+1\right)$ can be expressed as follows:(25)$$\hat{w}\left(k+1\right)=E_{w}\hat{w}\left(k\right)+E_{r}y_{p}\left(k\right)$$
where $E_{w}$ is the road input roll matrix, $E_{r}$ is the road input update matrix, and $y_{p}\left(k\right)$ is the previewed road height information at $N_{p}+1$ in front of the vehicle at moment k. In addition,
$$\hat{w}\left(k\right)=\left[\begin{array}{c} w\left(k\right)\\ w\left(k+1\right)\\ \vdots \\ w\left({k+N}_{p}\right) \end{array}\right],\;\hat{w}\left(k+1\right)=\left[\begin{array}{c} w\left(k+1\right)\\ w\left(k+2\right)\\ \vdots \\ w\left({k+N}_{p}+1\right) \end{array}\right],\;E_{w}=\left[\begin{array}{cccc} 0 & 1 & \cdots & 0\\ 0 & 0 & \ddots & \vdots \\ \vdots & \vdots & \ddots & 1\\ 0 & 0 & \cdots & 0 \end{array}\right],\;E_{r}=\left[\begin{array}{c} 0\\ \vdots \\ 0\\ 1 \end{array}\right]$$

### 3.3. Design of MPC Control Algorithm for VVP

MPC is a control algorithm that is highly resistant to disturbances and can efficiently handle complex systems with input constraints. In solving control problems, MPC is generally analyzed through discrete systems. In addition, the design and analysis process of MPC control algorithms generally includes three steps: model prediction, rolling optimization, and feedback correction.

As shown in Figure 12, the process of model prediction is to predict the system state within a certain time domain interval $\left[k,k+N_{p}\right]$ by using a system dynamic model based on the known system state and disturbance inputs. The controller calculates the inputs to the system in the entire time domain interval $\left[k,k+N_{p}\right]$ based on the predicted system response, but only the first solution is applied to the system. After the control input is applied to the system, the controller corrects the predicted system state according to the actual system state. At the next moment, the process is repeated to achieve rolling optimization.

Discrete systems are generally used to solve control problems with MPC control algorithms. Therefore, the system state Equation (9) is first discretized by zero-order holding input, fixed sampling time [35], and inverse Laplace transform. The system state equation after discretization for the combined vehicle and VVP can be expressed as
(26)$$\begin{array}{c} x_{d}\left(k+1\right)=A_{d}x_{d}\left(k\right)+B_{d}u\left(k\right)+E_{d}w\left(k\right)\\ y\left(k\right)=C_{d}x\left(k\right) \end{array}$$
where
$$A_{d}=e^{AT}=L^{-1}{\left(sI-A\right)}^{-1},\;B_{d}=\int_{0}^{T}e^{At}Bdt,\;E_{d}=\int_{0}^{T}e^{At}Edt,\;C_{d}=C$$
where T is the discrete period.

In order to improve the control performance of the VVP control algorithm, the original system matrix is augmented according to the road preview equation, and the augmented system state matrix is expressed as follows:(27)$$\begin{array}{c} x={\left[z_{1},z_{2},z_{3},z_{4},z_{c},z_{p},\varphi ,\theta ,{\dot{z}}_{1},{\dot{z}}_{2},{\dot{z}}_{3},{\dot{z}}_{4},{\dot{z}}_{c},{\dot{z}}_{p},\dot{\varphi },\dot{\theta },\hat{w}\right]}^{T}\\ y={\left[z_{p}-z_{c},{\ddot{z}}_{p}\right]}^{T} \end{array}$$

Combining Equations (25)–(27), the augmented discrete state space equation is expressed as
(28)$$\begin{array}{c} x\left(k+1\right)=\left[\begin{array}{ccc} A_{d} & E_{d} & 0\\ 0 & 0 & E_{w} \end{array}\right]x\left(k\right)+\left[\begin{array}{c} B_{d}\\ 0 \end{array}\right]u\left(k\right)+\left[\begin{array}{c} 0\\ E_{r} \end{array}\right]y_{p}\left(k\right)\\ y\left(k\right)=\left[\begin{array}{cc} C_{d} & 0 \end{array}\right]x\left(k\right) \end{array}$$

The generalized system state space equation can be simplified as
(29)$$\begin{array}{c} x\left(k+1\right)=A_{d,s}x\left(k\right)+B_{d,s}u\left(k\right)+E_{r,s}y_{p}\left(k\right)\\ y\left(k\right)=C_{d,s}x\left(k\right) \end{array}$$
where
$$A_{d,s}=\left[\begin{array}{ccc} A_{d} & E_{d} & 0\\ 0 & 0 & E_{w} \end{array}\right],\;B_{d,s}=\left[\begin{array}{c} B_{d}\\ 0 \end{array}\right],\;E_{r,s}=\left[\begin{array}{c} 0\\ E_{r} \end{array}\right],\;C_{d,s}=\left[\begin{array}{cc} C_{d} & 0 \end{array}\right]$$

The MPC vibration isolation platform control algorithm based on road preview is solved by converting the control system cost function into a quadratic programming form. The general form of quadratic programming [36] is expressed as
(30)$$\min\left(\frac{1}{2}y^{T}Qy+C^{T}y\right)$$

In the design of the control algorithm of the VVP, the purpose is to improve the vibration isolation performance of the VVP. It is required that the vibration of the VVP is as small as possible. Therefore, the cost function of the system at moment k can be expressed as
(31)$$\begin{array}{l} J=\frac{1}{2}\sum_{i=0}^{N_{p}-1}\left[y^{T}\left(k+i|k\right)Qy\left(k+i|k\right)+u^{T}\left(k+i|k\right)Ru\left(k+i|k\right)\right]+\\ \;\;\;\;\;\;\;\;y^{T}\left(k+N_{p}|k\right)Sy\left(k+N_{p}|k\right) \end{array}$$
where Q, R, and S are the weights of error, input, and terminal errors, respectively. $y\left(k+i|k\right)$ and $u\left(k+i|k\right)$ are the values predicted at moment k for y and u at moment k+i, respectively.

At moment k, the predicted values of the system state x according to the state space equation of the system after augmentation are expressed as
(32)$$\begin{array}{l} x\left(k|k\right)=x\left(k\right)\\ x\left(k+1|k\right)=A_{d,s}x\left(k\right)+B_{d,s}u\left(k|k\right)+E_{r,s}y_{p}\left(k\right)\\ \vdots \\ x\left(k+N_{p}|k\right)=A_{d,s}^{N_{p}}x\left(k\right)+A_{d,s}^{N_{p}-1}B_{d,s}u\left(k|k\right)+,\cdots \cdots ,+B_{d,s}u\left(k+N_{p}-1|k\right)+\\ \;\;\;\;\;\;\;\;A_{d,s}^{N_{p}-1}E_{r,s}y_{p}\left(k\right)+,\cdots \cdots ,+E_{r,s}y_{p}\left(k+N_{p}-1\right) \end{array}$$
which can be simplified as
(33)$$X\left(k\right)={\bar{A}}_{d,s}x\left(k\right)+{\bar{B}}_{d,s}U\left(k\right)+{\bar{E}}_{r,s}Y_{p}\left(k\right)$$
where
$$X\left(k\right)=\left[\begin{array}{c} x\left(k|k\right)\\ x\left(k+1|k\right)\\ \vdots \\ x\left(k+N_{p}|k\right) \end{array}\right],\;U\left(k\right)=\left[\begin{array}{c} u\left(k|k\right)\\ u\left(k+1|k\right)\\ \vdots \\ u\left(k+N_{p}-1|k\right) \end{array}\right],\;\;Y_{p}\left(k\right)=\left[\begin{array}{c} y_{p}\left(k\right)\\ y_{p}\left(k+1\right)\\ \vdots \\ y_{p}\left(k+N_{p}-1\right) \end{array}\right],\;{\bar{A}}_{d,s}=\left[\begin{array}{c} I\\ A_{d,s}\\ \vdots \\ A_{d,s}^{N_{p}} \end{array}\right],\;\;{\bar{B}}_{d,s}=\left[\begin{array}{cccc} 0 & 0 & \cdots & 0\\ B_{d,s} & 0 & \cdots & 0\\ A_{d,s}B_{d,s} & B_{d,s} & \ddots & \vdots \\ \vdots & \ddots & \ddots & 0\\ A_{d,s}^{N_{p}-1}B_{d,s} & \cdots & A_{d,s}B_{d,s} & B_{d,s} \end{array}\right],\;{\bar{E}}_{r,s}=\left[\begin{array}{cccc} 0 & 0 & \cdots & 0\\ E_{r,s} & 0 & \cdots & 0\\ A_{d,s}E_{r,s} & E_{r,s} & \ddots & \vdots \\ \vdots & \ddots & \ddots & 0\\ A_{d,s}^{N_{p}-1}E_{r,s} & \cdots & A_{d,s}E_{r,s} & E_{r,s} \end{array}\right]$$

In addition, Equation (31) can be simplified by the matrix form as
(34)$$J=\frac{1}{2}X^{T}\left(k\right){\bar{C}}_{d,s}^{T}\bar{Q}{\bar{C}}_{d,s}X\left(k\right)+\frac{1}{2}U^{T}\left(k\right)\bar{R}U\left(k\right)$$
where
$${\bar{C}}_{d,s}=\left[\begin{array}{cccc} C_{d,s} & & & \\ & \ddots & & \\ & & C_{d,s} & \\ & & & C_{d,s} \end{array}\right],\;\bar{Q}=\left[\begin{array}{cccc} Q & & & \\ & \ddots & & \\ & & Q & \\ & & & S \end{array}\right],\;\bar{R}=\left[\begin{array}{cccc} R & & & \\ & \ddots & & \\ & & R & \\ & & & R \end{array}\right]$$

By substituting Equation (33) into (34), the cost function of the quadratic form is obtained after simplification, as follows:(35)$$J=\frac{1}{2}U^{T}\left(k\right)Q_{q,s}U\left(k\right)+C_{c,s}^{T}U\left(k\right)+C_{e}$$
where
$$Q_{q,s}={\bar{B}}_{d,s}^{T}{\bar{C}}_{d,s}^{T}\bar{Q}{\bar{C}}_{d,s}{\bar{B}}_{d,s}+\bar{R}\;C_{c,s}={\bar{B}}_{d,s}^{T}{\bar{C}}_{d,s}^{T}\bar{Q}{\bar{C}}_{d,s}{\bar{A}}_{d,s}x\left(k\right)+{\bar{B}}_{d,s}^{T}{\bar{C}}_{d,s}^{T}\bar{Q}{\bar{C}}_{d,s}{\bar{E}}_{r,s}Y_{p}\left(k\right)\;C_{e}={\left[{\bar{A}}_{d,s}x\left(k\right)+{\bar{E}}_{r,s}Y_{p}\left(k\right)\right]}^{T}{\bar{C}}_{d,s}^{T}\bar{Q}{\bar{C}}_{d,s}\left[{\bar{A}}_{d,s}x\left(k\right)+{\bar{E}}_{r,s}Y_{p}\left(k\right)\right]$$

Meanwhile, considering the nonlinear constraint of the CDC damper (24), $U\left(k\right)$ is solved by the Quardprog function in MATLAB according to the cost function of the quadratic programming form, and the first set of solutions of $U\left(k\right)$ is applied to the controlled object.

The MPC is designed with road preview information and state prediction information. The MPC is solved by quadratic programming without approximation when calculating the control input force. The LQR is unable to add the road preview information and state prediction information. The LQR is solved by the solution of the Riccati approximation when calculating the control input force. Therefore, the control performance of MPC may be better.

## 4. Simulation and Test

Based on the proposed MPC control algorithm with road preview for VVP, relevant simulations and real tests are carried out, as specified in this section. In the simulation, the 8-DOF VVP simulation model and control algorithms are built in the MATLAB/Simulink environment. When the vehicle is driven on C-class random road surface, discrete impact road surface, and sinusoidal road surface, the vibration control of the VVP is simulated and analyzed. The vertical accelerations of the upper plane and base of the VVP are obtained. The acceleration of the upper plane and the base describes the vertical acceleration of the upper plane of the VVP and the connection point between the VVP and the body, respectively. The improvement of the vibration isolation performance by the proposed MPC control algorithm is compared and analyzed with a LQR control algorithm. In the test, the CDC shock absorber is controlled using a vehicle control unit (VCU) equipped with the control algorithm, and the vibration signals are acquired through a host computer. Tests and analyses are conducted when the vehicle is subjected to the sinusoidal road disturbances. For a clearer and more accurate analysis of the vibration isolation performance of VVP in the test, the vertical accelerations of the upper plane are obtained by sensors. The vibration response of the system controlled by the MPC control algorithm is compared with that of the passive state through the acceleration transfer rate.

### 4.1. Simulation Analysis of VVP Control Algorithm

In this section, the vibration response of the VVP is simulated and analyzed when the vehicle is driven on the three types of road surfaces. Meanwhile, based on the root mean square value (RMS), variance, maximum value, and minimum value of the vibration response of the VVP, the improvement in performance of the MPC control algorithm and LQR control algorithm are compared and analyzed.

#### 4.1.1. Simulation Analysis of Vibration Response of VVP under C-Class Random Road Disturbances

When the vehicle is driven at 20 km/h on the C-class random road surface, the vibration responses of the VVP controlled by the MPC and LQR are determined, as shown in Figure 13. Both the MPC control algorithm and LQR control algorithm can effectively improve the vibration isolation performance of the VVP under C-class random road disturbances compared with the passive state. In addition, the vertical acceleration of the upper plane under MPC control is smaller than that under LQR control. Due to the effect of the reaction forces between the car base and loaded platform and the platform dynamic deflection, the vertical acceleration of the base is slightly larger than that of LQR. The vibration response curves are analyzed according to statistics as shown in Table 2. The results show that the designed MPC control algorithm can effectively improve the vibration isolation performance of the VVP under C-class random road disturbances. Compared to the passive state, the improvement in vibration isolation performance is about 40% better than that of LQR.

#### 4.1.2. Simulation Analysis of Vibration Response of VVP under Discrete Impact Road Disturbances

In this section, the vibration response of the VVP is simulated and analyzed when the vehicle is driven on the impulse road surface and step road surface. Meanwhile, based on the RMS value, variance, maximum value, and minimum value of the vibration response of the VVP, the improvement in performance of the MPC control algorithm and LQR control algorithm are compared and analyzed.

When the vehicle is driven at 20 km/h on the impulse road surface, the vibration responses of the VVP controlled by the MPC and LQR are determined, as shown in Figure 14. Both the MPC control algorithm and the LQR control algorithm can effectively improve the vibration isolation performance of the VVP under impulse road disturbances compared with the passive state. The vibration performance under MPC and LQR control for impulse road disturbances is similar to that for C-class random road disturbances. The crest factor of the vibration response exceeds nine for some components, including the upper plane under MPC, the base under MPC, and the base under LQR. Therefore, a secondary evaluation method, Vibration Dose Value (VDV) [37], which focuses more on peak vibration, is introduced to evaluate the vibration isolation performance. The vibration response curves are analyzed according to the statistics as shown in Table 3; the results show that the designed MPC control algorithm can effectively improve the vibration isolation performance of the VVP under impulse road disturbances. Compared to the passive state, RMS and VDV, which represent vibration isolation performance, are about 40% and 27% better than LQR, respectively.

When the vehicle is driven at 20 km/h on the step road surface, the vibration response of the VVP controlled by the MPC and LQR is determined, as shown in Figure 15. Both the MPC control algorithm and the LQR control algorithm can effectively improve the vibration isolation performance of the VVP under step road disturbances compared with the passive state. The vibration performance under MPC and LQR control for step road disturbances is similar to that for C-class random road disturbances. The vibration response curves are analyzed according to the statistics as shown in Table 4; the results show that the designed MPC control algorithm can effectively improve the vibration isolation performance of the VVP under step road disturbances. Compared to the passive state, the improvement in vibration isolation performance is about 40% better than that of LQR.

#### 4.1.3. Simulation Analysis of Vibration Response of VVP under Sinusoidal Road Disturbances

In this section, the vibration response of the VVP is simulated and analyzed when the vehicle is driven on the sinusoidal fixed frequency road surface and sinusoidal swept frequency road surface. Meanwhile, based on the RMS value, variance, maximum value, and minimum value of the vibration response of the VVP, the improvement in performance of the MPC control algorithm and LQR control algorithm are compared and analyzed.

When all four wheels are subjected to the same sinusoidal fixed frequency disturbance, the vibration response of the VVP controlled by the MPC and LQR is determined, as shown in Figure 16. Both the MPC control algorithm and the LQR control algorithm can effectively improve the vibration isolation performance of the VVP under sinusoidal fixed frequency road disturbances compared with the passive state. In addition, the vertical acceleration of the upper plane under MPC control is smaller than that under LQR control. Due to the influence of road disturbance frequency, the vertical acceleration of the base is slightly smaller than that of LQR. The vibration response curves are analyzed according to the statistics as shown in Table 5; the results show that the designed MPC control algorithm can effectively improve the vibration isolation performance of the VVP under sinusoidal fixed frequency road disturbances. Compared to the passive state, the improvement in vibration isolation performance is about 30% better than that of LQR.

When all four wheels are subjected to the same sinusoidal swept frequency disturbance, the vibration response of the VVP controlled by the MPC and LQR is determined, as shown in Figure 17. Both the MPC control algorithm and the LQR control algorithm can effectively improve the vibration isolation performance of the VVP under sinusoidal swept frequency road disturbances compared with the passive state. The vibration performance under MPC and LQR control for sinusoidal swept frequency road disturbances is similar to that for C-class random road disturbances. The vibration response curves are analyzed according to the statistics as shown in Table 6; the results show that the designed MPC control algorithm can effectively improve the vibration isolation performance of the VVP under sinusoidal swept frequency road disturbances. Compared to the passive state, the improvement in vibration isolation performance is about 70% better than that of LQR.

According to the above analysis and calculations, as shown in Figure 18, the designed MPC control algorithm on a variety of road surfaces optimizes the RMS value of vertical acceleration of the VVP at about 80%, with small variation, which can effectively improve the vibration isolation performance of the VVP. The control performance of MPC is about 40% better than that of LQR. In addition, compared to other road surfaces, the optimization of the vibration response of the VVP under LQR control with sinusoidal swept frequency road disturbances is substantially reduced, by about 30%. This also shows that the variation of VVP vibration isolation performance with road surface under MPC control is smaller compared to that under LQR control, reflecting the stronger anti-interference performance of MPC.

### 4.2. Test Analysis of VVP Control Algorithm

In this test, the VVP is installed in a medium-sized compartment, as shown in Figure 19, and the vibration environment test under various working conditions is conducted using the MTS excitation table. The vibration isolation performance of the platform and the effectiveness of the MPC control algorithm based on road preview are verified by frequency domain and time domain analysis of the vertical acceleration data collected by sensors from the VVP.

In this section, vibration isolation tests of the VVP are conducted on sinusoidal swept frequency road disturbances and C-class random road disturbances. Through the tests on sinusoidal road disturbance, the resonance peak and effective vibration isolation frequency band of the VVP system can be obtained. The C-class random road disturbances test reflects the vibration of the equipment placed on the VVP when the vehicle is driven on a real road surface. Meanwhile, the vibration isolation performance of the VVP is compared and analyzed using parameters such as the RMS value of acceleration.

When all four wheels are subjected to the same sinusoidal swept frequency disturbance, the frequency domain vibration response of the VVP system controlled by the MPC is determined, as shown in Figure 20. At 2.25 Hz, the response curve of the vertical acceleration of the upper plane of the MPC-controlled VVP system reaches its peak, and this frequency is the resonant frequency of the MPC-controlled VVP system. At 2 Hz, the response curve of the vertical acceleration of the upper plane of the VVP system in the passive state reaches its peak, and this frequency is the resonant frequency of the VVP system in the passive state. The vibration of the upper platform under MPC control can be effectively suppressed in the low and resonant frequency bands. In addition, the real model of VVP mounted on the car is a multi-DOF model with multiple resonance peaks. With the frequency increasing at around 72 Hz, the response curve reaches the second resonance peak. The vibration response curves are analyzed according to the statistics as shown in Table 7; the results show that the designed MPC control algorithm can effectively improve the vibration isolation performance of the VVP under sinusoidal swept frequency road disturbances, and the improvement in vibration isolation performance is about 21% better than that of the passive state.

When the vehicle is driven at 20 km/h on the C-class random road disturbances, the vibration response of the VVP system controlled by the MPC is determined, as shown in Figure 21. The vertical acceleration of the VVP is reduced relative to the base. The vibration response under MPC control is improved compared to the passive state. The vibration response curves are analyzed according to the statistics as shown in Table 8; the results show that the designed MPC control algorithm can improve the vibration isolation performance under C-class random road disturbances, and the improvement in vibration isolation performance is about 15% better than that of the passive state.

According to the simulation and test results, the test results are inferior to the simulation results in terms of the optimization of each index. The main reasons for the above phenomenon are as follows:(1)The test vehicle is modified several times to change the vibration environment of the vehicle itself, in addition to considering the nonlinearity of the CDC shock absorbers in the controller design. As a result, the control performance of the controller designed according to the original vehicle parameters is reduced.(2)The vibration of the VVP and mounted vehicle is affected by vibration of internal components of the vehicle. The test conditions are worse than the simulation conditions. Therefore, in the real tests, the vibration response under MPC control is slightly worse than that under the passive state at some frequencies.(3)The damping of the CDC damper of the VVP in the passive state is the optimal value determined through several tests; therefore the vibration isolation performance of the VVP in the passive state is also better.

## 5. Conclusions

In this paper, to improve the vibration isolation performance of the platform by realizing the control of the VVP, mathematical modeling combining the delivery vehicle and the isolation platform is first carried out. The CDC damper is modeled based on the external characteristics test. The accurate damping force, velocity, and current variation relationships of the CDC damper are obtained. The map graph is transformed into a set of inequalities as an actuator nonlinear constraint. According to the actuator constraints and road preview information, the MPC control algorithm is designed to meet the VVP control requirements. Simulation results show that the MPC control algorithm is about 40% better than the LQR control algorithm in optimizing the root mean square value of the platform vertical acceleration on different road surfaces. The test results show that the VVP under MPC control shows improvement in vibration isolation performance compared to the passive systems. Therefore, the designed MPC control algorithm based on road preview has significant potential for platform vibration isolation control and can effectively improve the stability of the equipment placed on the VVP when the vehicle is driven on rough terrain.

Considering the structural nonlinearity of the VVP and the uncertainty of the road environment, a multi-physics simulation model will be built in future work. The MPC weighting matrix will be further optimized using a learning-based algorithm in the control algorithm design. In future tests, sensors such as a binocular camera will be used to obtain road surface information, and road tests will be conducted.

## Figures and Tables

**Figure 1 sensors-24-00243-f001:**
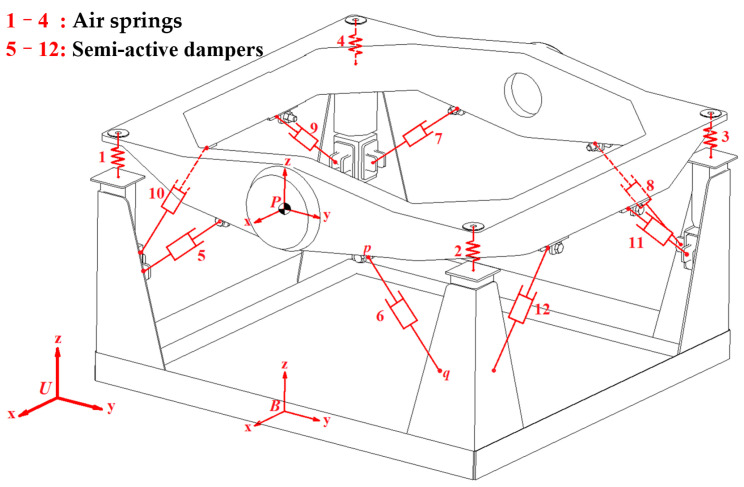
A VVP model.

**Figure 2 sensors-24-00243-f002:**
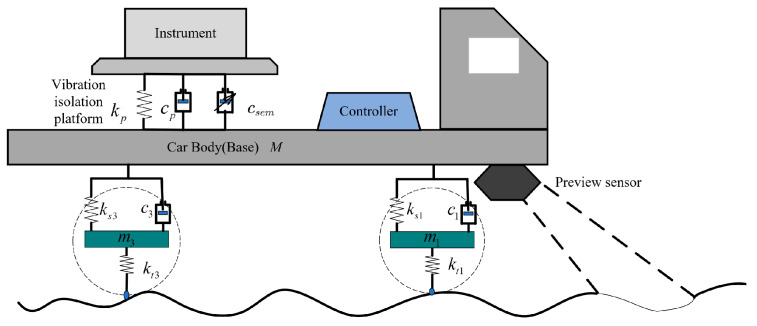
A model based on road preview, combining a VVP with a mounted vehicle.

**Figure 3 sensors-24-00243-f003:**
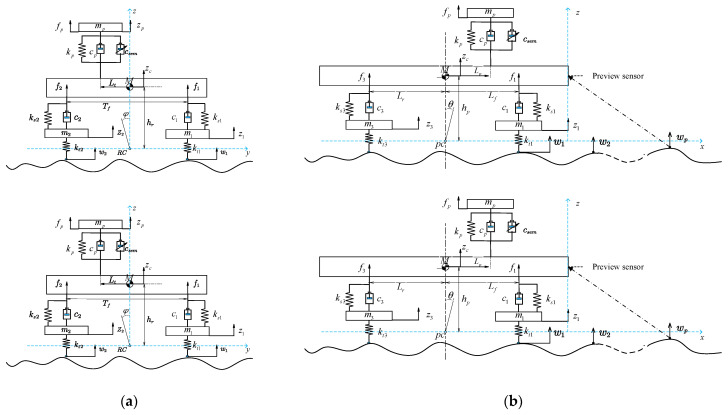
Combined vehicle and VVP model: (**a**) front view of the platform model; (**b**) left view of the platform model.

**Figure 4 sensors-24-00243-f004:**
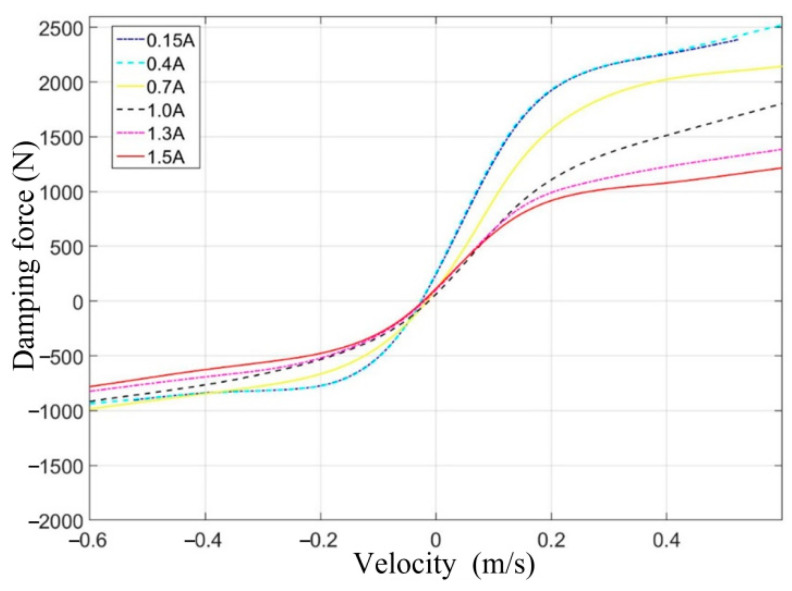
The relationship curves between damping force, velocity, and current.

**Figure 5 sensors-24-00243-f005:**
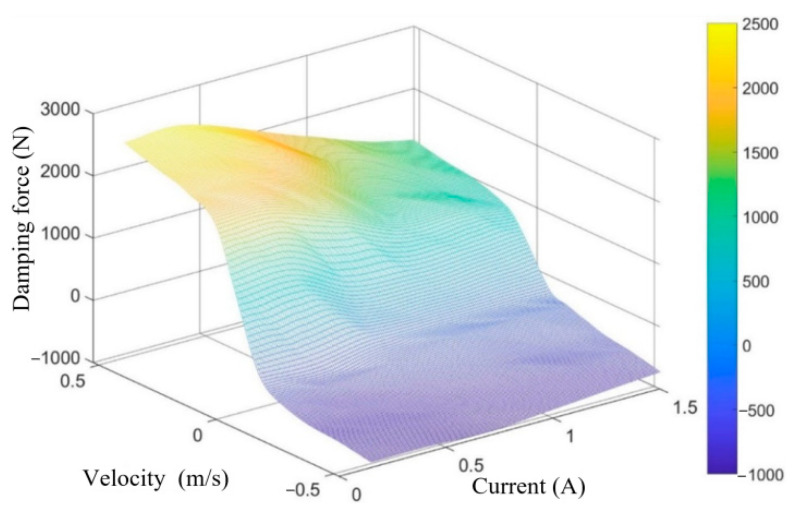
Three-dimensional MAP diagram of damping force, velocity, and current.

**Figure 6 sensors-24-00243-f006:**
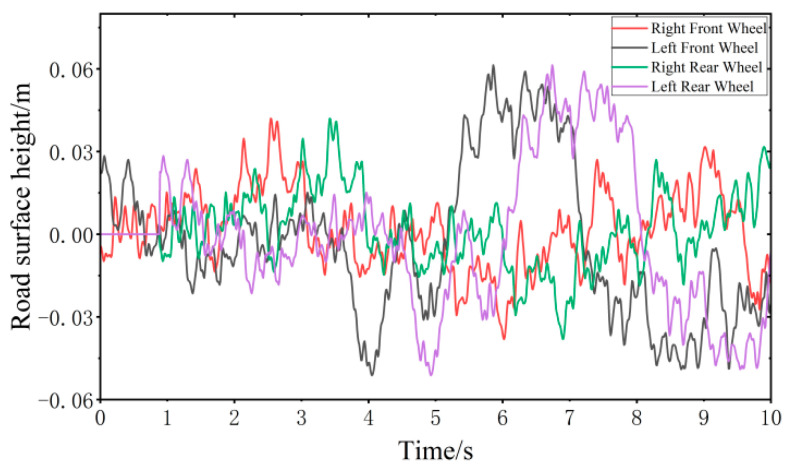
C-class random road surface performance model.

**Figure 7 sensors-24-00243-f007:**
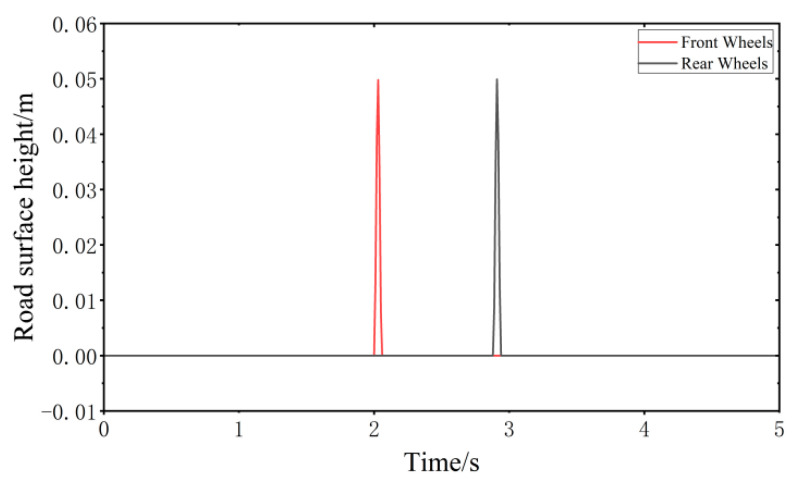
Impulse road surface performance model.

**Figure 8 sensors-24-00243-f008:**
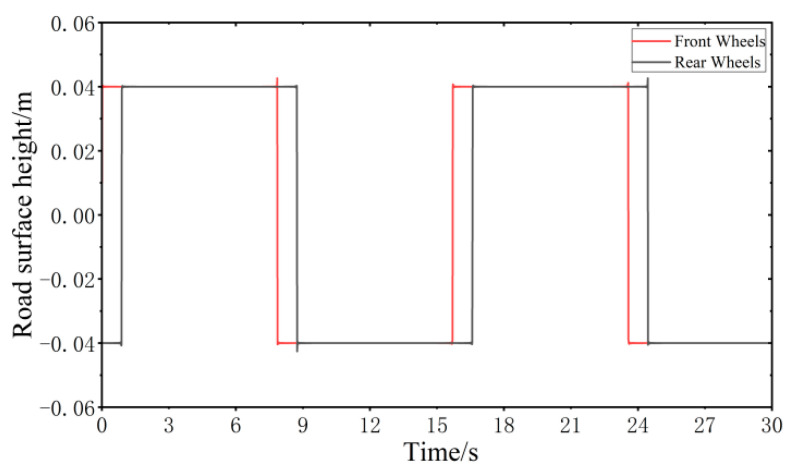
Step road surface performance model.

**Figure 9 sensors-24-00243-f009:**
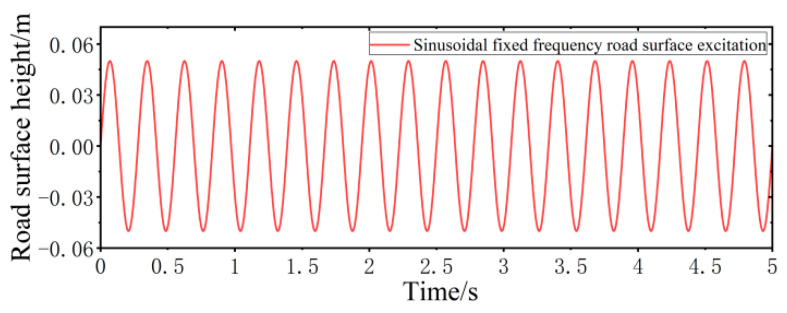
Sinusoidal fixed frequency road surface performance model.

**Figure 10 sensors-24-00243-f010:**
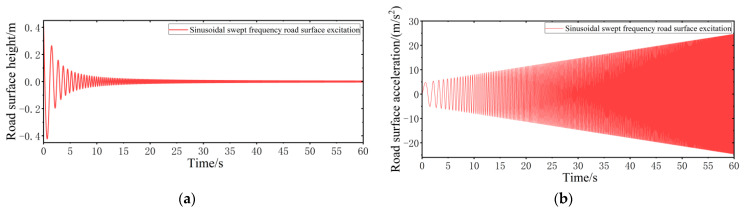
Sinusoidal swept frequency road surface performance model: (**a**) road surface height input; (**b**) road surface acceleration input.

**Figure 11 sensors-24-00243-f011:**
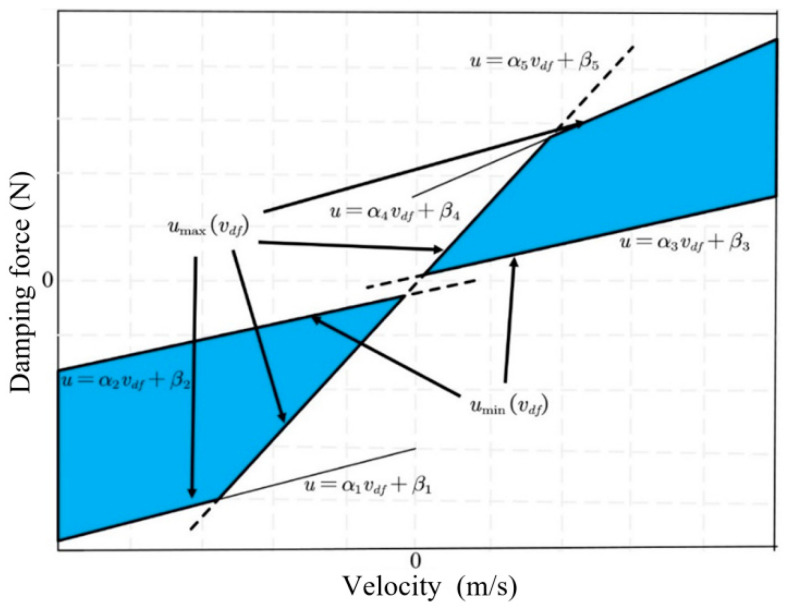
CDC damper damping force constraint interval.

**Figure 12 sensors-24-00243-f012:**
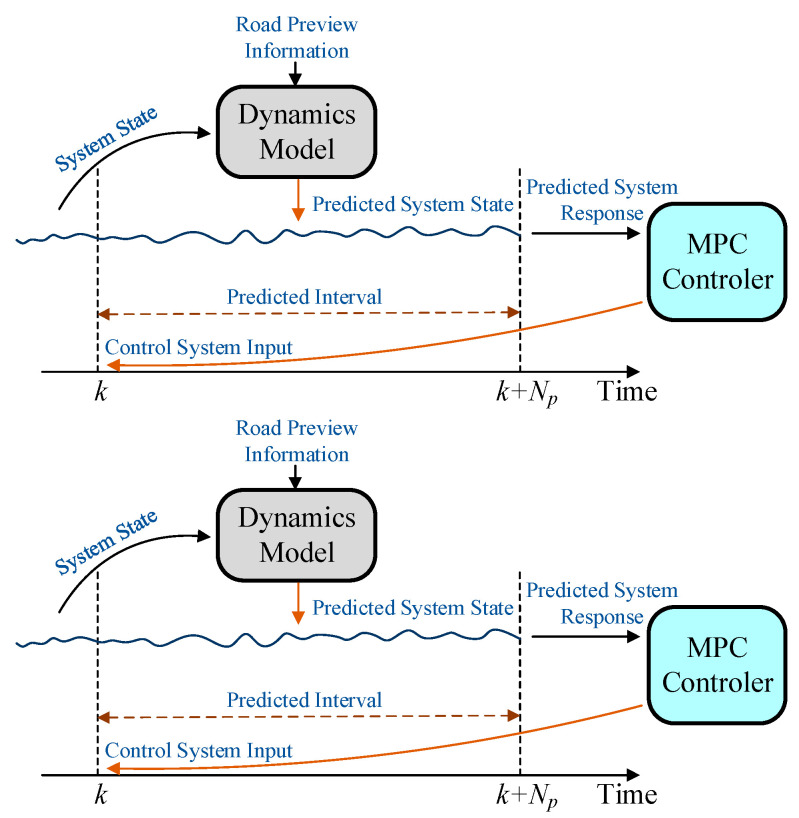
The design and analysis process of MPC control algorithms.

**Figure 13 sensors-24-00243-f013:**
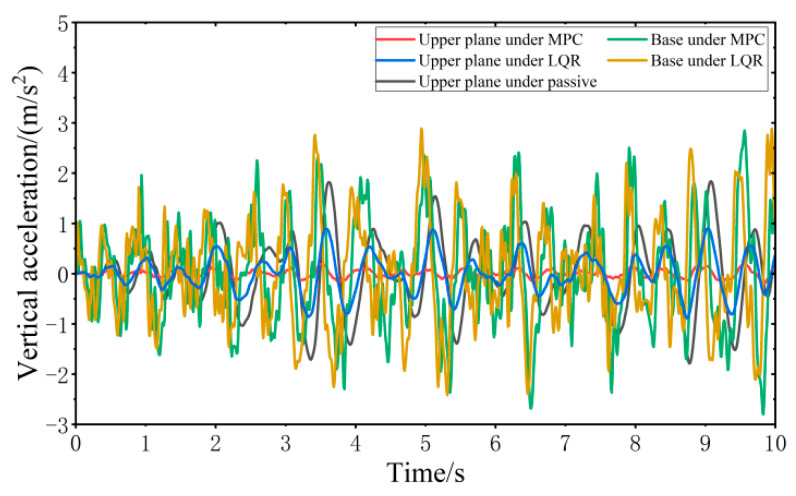
Vibration response of VVP under C-class random road disturbances.

**Figure 14 sensors-24-00243-f014:**
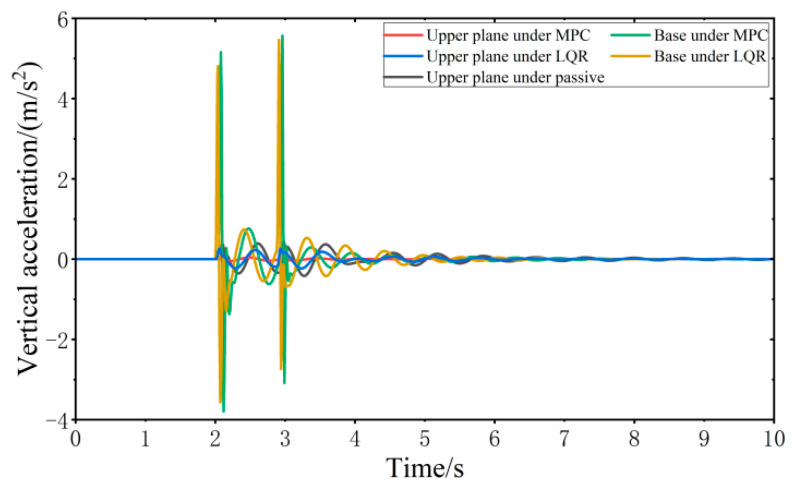
Vibration response of VVP under impulse road disturbances.

**Figure 15 sensors-24-00243-f015:**
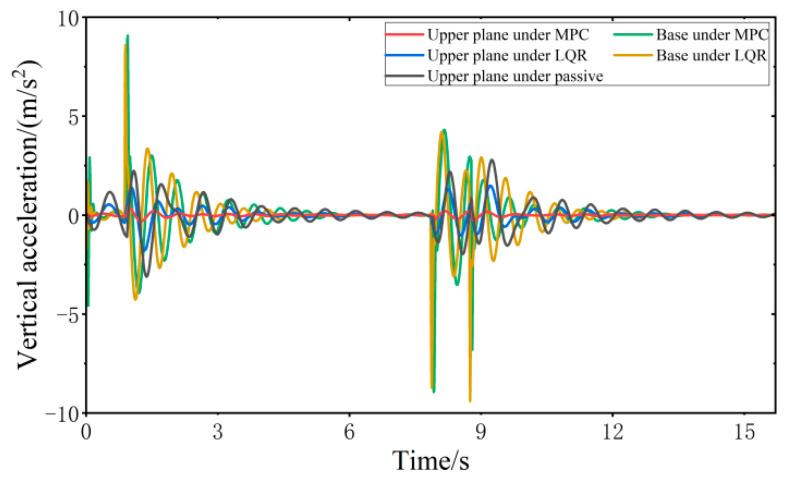
Vibration response of VVP under step road disturbances.

**Figure 16 sensors-24-00243-f016:**
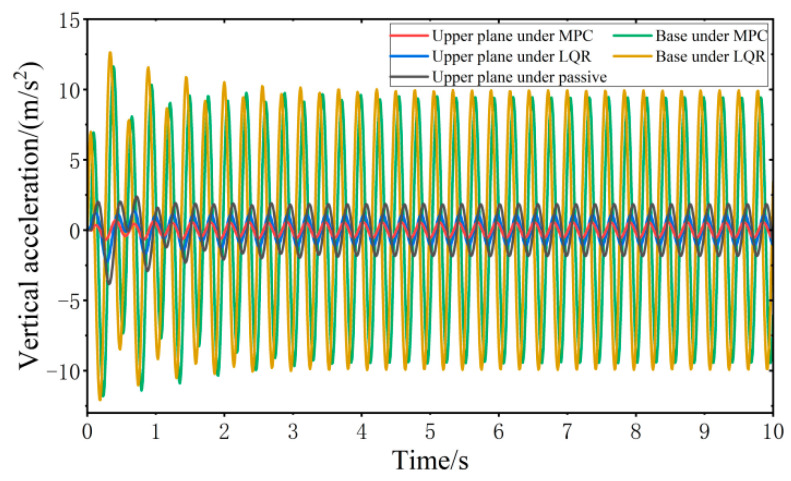
Vibration response of VVP under sinusoidal fixed frequency road disturbances.

**Figure 17 sensors-24-00243-f017:**
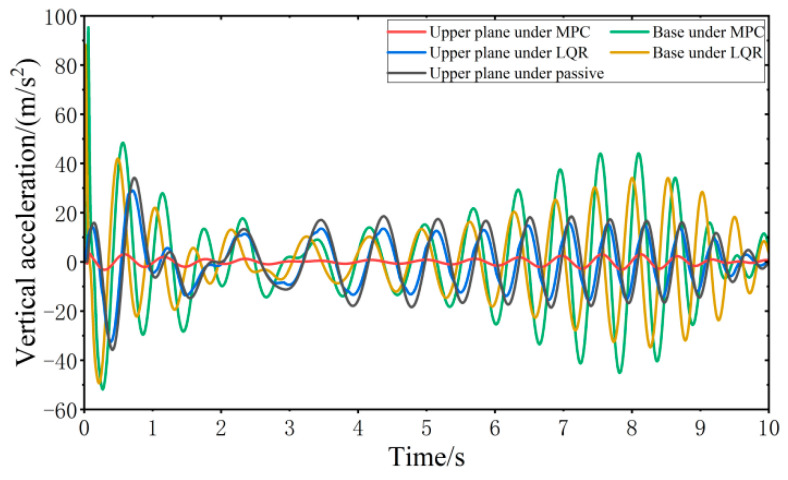
Vibration response of VVP under sinusoidal swept frequency road disturbances.

**Figure 18 sensors-24-00243-f018:**
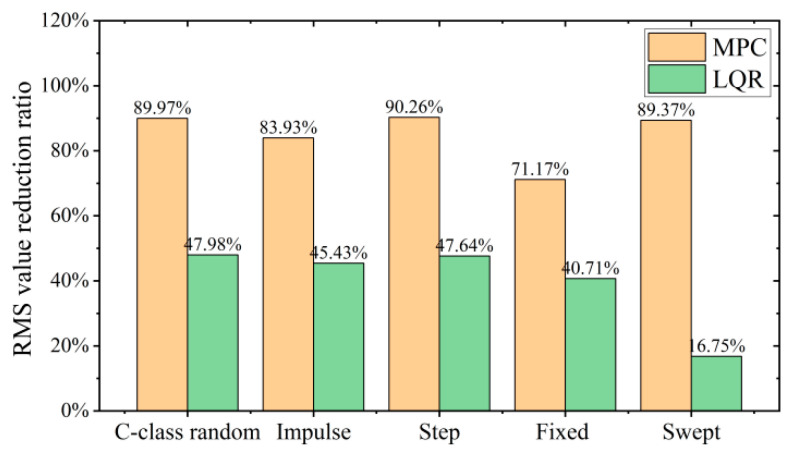
RMS value reduction ratio of the MPC and the LQR for different road surfaces.

**Figure 19 sensors-24-00243-f019:**
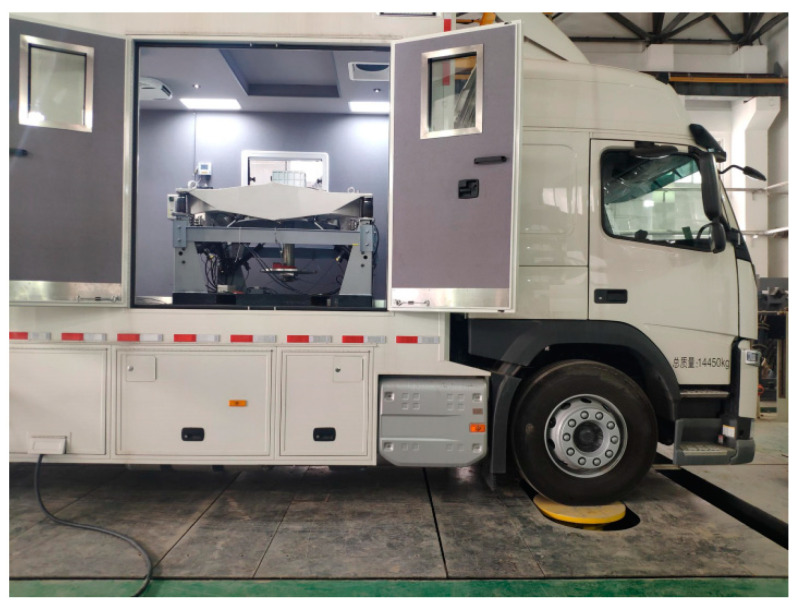
Real model of VVP mounted on the car and MTS excitation table.

**Figure 20 sensors-24-00243-f020:**
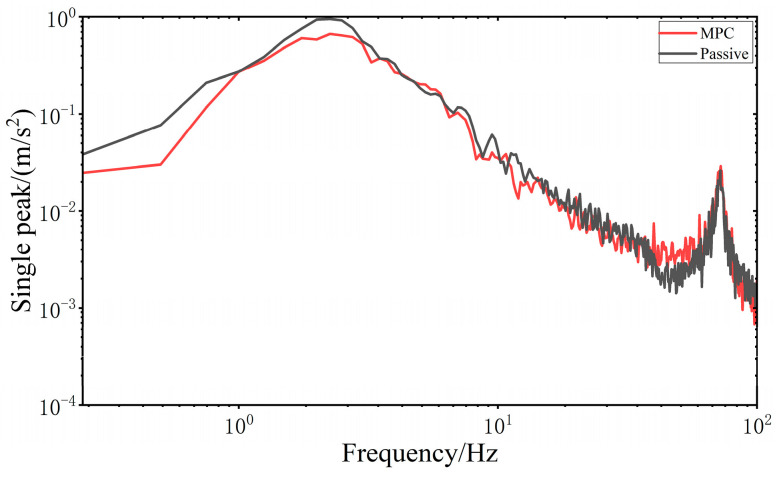
Frequency domain vibration response of the upper plane under sinusoidal swept frequency road disturbances.

**Figure 21 sensors-24-00243-f021:**
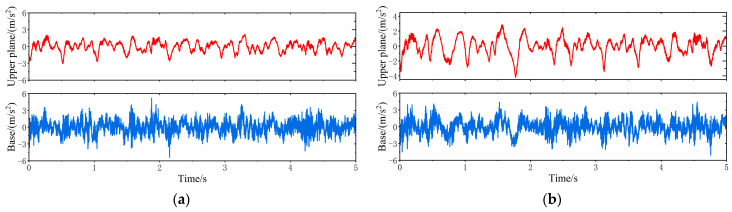
Time domain vibration response of the VVP under C-class random road disturbances: (**a**) MPC test results; (**b**) passive test results.

**Table 1 sensors-24-00243-t001:** Parameters of VVP and vehicle.

Parameter	Value
VVP mass $m_{p}$	1000 kg
Stiffness of VVP spring $k_{p}$	40,000 N/m
Damping coefficients of VVP damper $c_{p}$	3500 N·s/m
Damping coefficients of semi-active CDC damper $c_{sem}$	1000–3500 N·s/m
Car body mass M	10,000 kg
Car body pitch moment of inertia $I_{p}$	95,550 $kg\cdot m^{2}$
Car body roll moment of inertia $I_{r}$	13,650 $kg\cdot m^{2}$
Stiffness of four suspension springs $k_{si}$	500,000 N/m
Damping coefficients of four suspension dampers $c_{i}$	10,000 N·s/m
Distance from the mass center of the car body to the roll center $h_{r}$	1 m
Distance from the mass center of the car body to the pitch center $h_{p}$	1.2 m
Lateral distances from the mass center of the VVP to the mass center of the car body $L_{t}$	0.45 m
Longitudinal distances from the mass center of the VVP to the mass center of the car body $L_{p}$	1.2 m
Front axle wheelbase $T_{f}$	2.035 m
Rear axle wheelbase $T_{r}$	1.834 m
Horizontal distances from the front axles to the mass center of the car body $L_{f}$	2.4 m
Horizontal distances from the rear axles to the mass center of the car body $L_{r}$	2.5 m
Unsprung masses $m_{1}$, $m_{2}$	227 kg
Unsprung masses $m_{3}$, $m_{4}$	105 kg
Stiffness of four wheels $k_{ti}$	900,000 N/m
Gravitational acceleration g	9.8 $m/s^{2}$

**Table 2 sensors-24-00243-t002:** Simulation results of vibration response of VVP under C-class random road disturbances.

	Parameter	$\mathrm{RMS}\;m/s^{2}$	$\mathrm{Var}\;{\left(m/s^{2}\right)}^{2}$	$\mathrm{Max}\;m/s^{2}$	$\mathrm{Min}\;m/s^{2}$
MPC	Upper plane	0.0718	0.0052	0.1945	−0.1940
	Base	1.0647	1.1348	2.8500	−2.7985
	Reduction Ratio	93.26%	99.54%	93.18%	93.07%
LQR	Upper plane	0.3725	0.1389	0.9000	−0.8868
	Base	1.0426	1.0878	2.8879	−2.4175
	Reduction Ratio	64.27%	87.23%	68.84%	63.32%
Passive	Upper plane	0.7161	0.5133	1.8383	−1.7863
Compared with Passive State	MPC Reduction Ratio	89.97%	98.99%	89.42%	89.14%
	LQR Reduction Ratio	47.98%	72.94%	51.04%	50.36%

**Table 3 sensors-24-00243-t003:** Simulation results of vibration response of VVP under impulse road disturbances.

	Parameter	$\mathrm{RMS}\;m/s^{2}$	$\mathrm{Var}\;{\left(m/s^{2}\right)}^{2}$	$\mathrm{Max}\;m/s^{2}$	$\mathrm{Min}\;m/s^{2}$
MPC	Upper plane	0.0190	0.0004	0.2791	−0.0661
	Base	0.4341	0.1886	5.5630	−3.7965
	Reduction Ratio	95.62%	99.79%	94.98%	98.26%
LQR	Upper plane	0.0645	0.0042	0.2615	−0.2314
	Base	0.4403	0.1940	5.4620	−3.5676
	Reduction Ratio	85.35%	97.84%	95.21%	93.51%
Passive	Upper plane	0.1182	0.0140	0.4553	−0.4195
Compared with Passive State	MPC Reduction Ratio	83.93%	97.14%	38.70%	84.24%
	LQR Reduction Ratio	45.43%	70.00%	42.57%	44.84%

**Table 4 sensors-24-00243-t004:** Simulation results of vibration response of VVP under step road disturbances.

	Parameter	$\mathrm{RMS}\;m/s^{2}$	$\mathrm{Var}\;{\left(m/s^{2}\right)}^{2}$	$\mathrm{Max}\;m/s^{2}$	$\mathrm{Min}\;m/s^{2}$
MPC	Upper plane	0.0713	0.0051	0.4302	−0.3100
	Base	1.0901	1.1890	9.0661	−8.9382
	Reduction Ratio	93.46%	99.57%	95.25%	96.53%
LQR	Upper plane	0.3833	0.1470	1.4854	−1.7899
	Base	1.1503	1.3240	8.5950	−9.3939
	Reduction Ratio	66.68%	88.90%	82.72%	80.95%
Passive	Upper plane	0.7320	0.5362	2.7883	−3.1152
Compared with Passive State	MPC Reduction Ratio	90.26%	99.05%	84.57%	90.05%
	LQR Reduction Ratio	47.64%	72.58%	46.73%	42.54%

**Table 5 sensors-24-00243-t005:** Simulation results of vibration response of VVP under sinusoidal fixed frequency road disturbances.

	Parameter	$\mathrm{RMS}\;m/s^{2}$	$\mathrm{Var}\;{\left(m/s^{2}\right)}^{2}$	$\mathrm{Max}\;m/s^{2}$	$\mathrm{Min}\;m/s^{2}$
MPC	Upper plane	0.3908	0.1529	0.6997	−0.6934
	Base	6.7090	45.0542	11.6264	−11.810
	Reduction Ratio	94.17%	99.66%	93.98%	94.13%
LQR	Upper plane	0.8038	0.6467	1.3555	−2.2407
	Base	7.0508	49.7615	12.6294	−12.078
	Reduction Ratio	88.60%	98.70%	89.27%	81.45%
Passive	Upper plane	1.3556	1.8394	2.3813	−3.8393
Compared with Passive State	MPC Reduction Ratio	71.17%	91.69%	70.62%	81.94%
	LQR Reduction Ratio	40.71%	64.84%	43.08%	41.64%

**Table 6 sensors-24-00243-t006:** Simulation results of vibration response of VVP under sinusoidal swept frequency road disturbances.

	Parameter	$\mathrm{RMS}\;m/s^{2}$	$\mathrm{Var}\;{\left(m/s^{2}\right)}^{2}$	$\mathrm{Max}\;m/s^{2}$	$\mathrm{Min}\;m/s^{2}$
MPC	Upper plane	1.3009	1.6941	3.3912	−3.2347
	Base	19.9942	400.159	95.3391	−51.885
	Reduction Ratio	93.49%	99.58%	96.44%	93.77%
LQR	Upper plane	10.1864	103.867	28.9921	−32.532
	Base	16.5445	273.993	88.1629	−49.382
	Reduction Ratio	38.43%	62.09%	67.12%	34.12%
Passive	Upper plane	12.2366	149.884	34.1697	−35.782
Compared with Passive State	MPC Reduction Ratio	89.37%	98.87%	90.08%	90.96%
	LQR Reduction Ratio	16.75%	30.70%	15.15%	9.08%

**Table 7 sensors-24-00243-t007:** Test results of vibration response under sinusoidal swept frequency road disturbances.

	Parameter	$\mathrm{RMS}\;m/s^{2}$	$\mathrm{Var}\;{\left(m/s^{2}\right)}^{2}$	$\mathrm{Max}\;m/s^{2}$	$\mathrm{Min}\;m/s^{2}$
Upper plane	MPC	0.0945	0.0082	0.6675	0.0007
	Passive	0.1202	0.0134	0.9521	0.0010
	Reduction Ratio	21.38%	38.81%	29.89%	30.00%

**Table 8 sensors-24-00243-t008:** Test results of vibration response under C-class random road disturbances.

	Parameter	$\mathrm{RMS}\;m/s^{2}$	$\mathrm{Var}\;{\left(m/s^{2}\right)}^{2}$	$\mathrm{Max}\;m/s^{2}$	$\mathrm{Min}\;m/s^{2}$
MPC	Upper plane	0.8987	0.8077	2.1851	−3.0522
	Base	1.1293	1.2756	5.1878	−5.3750
	Reduction Ratio	20.42%	36.68%	57.88%	43.22%
Passive	Upper plane	1.1627	1.3522	2.8669	−4.1945
	Base	1.2289	1.5104	4.3984	−5.0833
	Reduction Ratio	5.38%	10.47%	34.82%	17.49%

## Data Availability

Data are contained within the article.

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
