# Peer review of "Model Predictive Control of a Semi-Active Vehicle-Mounted Vibration Isolation Platform"

_sensors, 2023, doi:10.3390/s24010243_

Round 1

Reviewer 1 Report

Comments and Suggestions for Authors

The authors propose a structure and control method for an on-board vibration isolation device. This topic is interesting, but there are areas that need to be supplemented and modified. Here are some comments

(1) The theoretical contributions should be stressed in detail in Introduction

(2) Please check carefully all notations and equations. Moreover, the usage of English should be improved.

(3) Advantages of the proposed algorithm upon the well-known algorithms should be stressed. I suggest mentioning/comparing the simulations with the results of the recent related valid references.

(4) In introduction, it is not enough to state the current work. It should be expended and reconstructed. Including the motivation, the main difficulties, the main work and the improvements compared with previous related works should be emphasized in this section.

(5) The importance of the problem considered in this paper should be further addressed.

(6) The types of software employed for solving the problem and also simulation experiments should be stated clearly.

(7) The directions to further and improve the work should be added as future recommendation section after ‘conclusions’ section.

(8) I think the author should provide a more detailed description of the experimental scenarios and highlight their representativeness in practical applications.

(9) The authors should describe in more detail the structure and function of the "preview sensors" shown in Figure 1, emphasizing how the collected data are used.

(10) Whether the two road surface performance models formulated in 2.3.3 are consistent with the common road characteristics in practical applications? I believe that the use of the sinusoidal form is more regular and difficult to reflect the uncertainty of the road surface shape in practical applications.

(11) The authors should have gone into more detail about the novelty of the designed model predictive controller, which is not sufficiently innovative in the control methodology from the current version.

(12) A section should be added to the manuscript to illustrate the advantages of the proposed device and method by the change in the intensity of the vibration to which it is subjected when placed on a vibration isolation platform. This means that more attention should be paid to the effect of use and experience of the direct application target.

Author Response

Dear Reviewer,

I am pleased to resubmit the revised version of our manuscript, entitled “Model predictive control of a semi-active vehicle-mounted vibration isolation platform”.

I would like to thank you for the insightful comments and for taking the time and effort to provide the detailed guidance. Your comments are very helpful to improve our paper. We have revised the manuscript according to the reviewers’ comments, and the revised parts have been highlighted in red in the revised manuscript. Our responses to the reviewers' comments are placed in the attachment.

Best regards,

Weizhou Zhang

Reviewer 2 Report

Comments and Suggestions for Authors

The topic of intelligent suspensions in vehicles is very topical. They improve the comfort of transported people and the safety of transported goods. The suspensions of modern passenger cars and transport vehicles are now advanced mechatronic systems using the latest sensors and vision systems. 

The research plan was developed correctly. The simulation tests were performed correctly. The results of the simulation tests were well presented in the form of tables and graphically nice drawings. A very important achievement of the authors was the validation of previously performed simulation studies in real conditions. The simulation model was used in a real control system mounted on a truck. The researchers used an iterative testing approach to find the optimal values for each index.

Despite the high substantive value of the article, it must be supplemented before publication in the following areas:

1) The bibliography list contains 26 items. This is definitely not enough. The literature list contains too many older items from before 2010 (item 10 - 1997, item 11 - 1999, item 12 - 1997, item 17 - 2001, item 25 - 2002). Please check if there are any newer equivalents. The literature analysis includes too few articles written by European and American scientists. At least 20 consecutive recent scientific articles (published after 2020) should be analyzed.

2) The analysis contained in the Introduction is not sufficient. A thorough review of the technologies currently used in the suspension of luxury cars and transport vehicles should be undertaken. There should also be information about the types of innovative suspensions (mechatronic, pneumatic, hydraulic, mixed). Then, an overview of the sensors used in these suspensions and the methods of measuring and processing the signals should be carried out. In this regard, the 10 latest solutions described in European scientific articles should be analyzed.

3) The Introduction should include at least one drawing presenting a functional diagram of a modern vehicle suspension, including the sensors used, actuators, electronic control units and connections between them.

4) In the Summary and final conclusions, it should be emphasized that the use of predictive methods in the area of visual analysis of the road in front of the vehicle may have practical applications in currently manufactured vehicles. It should be indicated in what types of vehicles it can be used (luxury cars, transport vehicles, autonomous driving platforms).

5) The directions of continuation of initiated scientific research should also be described.

Author Response

(The authors gave the same response as above.)

Reviewer 3 Report

Comments and Suggestions for Authors

This manuscript presents a comprehensive research on the model predictive control of a semi-active vehicle-mounted vibration isolation platform. In the introduction section, the advantage of the proposed research method is not explained clearly. It seem that all the methods used in this manuscript are from the references, even some paramenters' values. Although lot of equations are listed in this manuuscript, most of them are from other references. Therefore, the novelty of this manuscript is not enough. 

In addition, the test just compares the MPC and passive mode under sinusoidal swept frequency road disturbances. This condition is much simpler than the simulation condition. The silumlation model is not verified.

Author Response

(The authors gave the same response as above.)

Round 2

Reviewer 1 Report

Comments and Suggestions for Authors

The author has made changes and additions and I have no further comments. I think it is acceptable in its current manuscript version.

Reviewer 3 Report

Comments and Suggestions for Authors

The authors have revised this manusript based on the reviewer's comments. This manuscript can be accepted now.